# Learning Robust Representations by Projecting Superficial Statistics Out

**Haohan Wang**
Carnegie Mellon University
Pittsburgh, PA USA
haohanw@cs.cmu.edu

**Zexue He**
Beijing Normal University
Beijing, China
zexueh@mail.bnu.edu.cn

**Zachary C. Lipton**
Carnegie Mellon University
Pittsburgh, PA, USA
zlipton@cmu.edu

**Eric P. Xing**
Carnegie Mellon University
Pittsburgh, PA, USA
epxing@cs.cmu.edu

## Abstract

Despite impressive performance as evaluated on i.i.d. holdout data, deep neural networks depend heavily on superficial statistics of the training data and are liable to break under distribution shift. For example, subtle changes to the background or texture of an image can break a seemingly powerful classifier. Building on previous work on domain generalization, we hope to produce a classifier that will generalize to previously unseen domains, even when domain identifiers are not available during training. This setting is challenging because the model may extract many distribution-specific (superficial) signals together with distribution-agnostic (semantic) signals. To overcome this challenge, we incorporate the *gray-level co-occurrence matrix* (GLCM) to extract patterns that our prior knowledge suggests are superficial: they are sensitive to texture but unable to capture the gestalt of an image. Then we introduce two techniques for improving our networks' out-of-sample performance. The first method is built on the reverse gradient method that pushes our model to learn representations from which the GLCM representation is not predictable. The second method is built on the independence introduced by projecting the model's representation onto the subspace orthogonal to GLCM representation's. We test our method on battery of standard domain generalization data sets and, interestingly, achieve comparable or better performance as compared to other domain generalization methods that explicitly require samples from the target distribution for training.

## 1 Introduction

Imagine training an image classifier to recognize facial expressions. In the training data, while all images labeled *"smile"* may actually depict smiling people, the "smile" label might *also* be correlated with other aspects of the image. For example, people might tend to smile more often while outdoors, and to frown more in airports. In the future, we might encounter photographs with previously unseen backgrounds, and thus we prefer models that rely as little as possible on the superficial signal.

The problem of learning classifiers robust to distribution shift, commonly called *Domain Adaptation (DA)*, has a rich history. Under restrictive assumptions, such as covariate shift (Shimodaira, 2000; Gretton et al., 2009), and label shift (also known as *target shift* or *prior probability shift*) (Storkey, 2009; Schölkopf et al., 2012; Zhang et al., 2013; Lipton et al., 2018), principled methods exist for estimating the shifts and retraining under the importance-weighted ERM framework. Other papers bound worst-case performance under bounded shifts as measured by divergence measures on the train v.s. test distributions (Ben-David et al., 2010a; Mansour et al., 2009; Hu et al., 2016).

While many impossibility results for DA have been proven (Ben-David et al., 2010b), humans nevertheless exhibit a remarkable ability to function out-of-sample, even when confronting dramatic

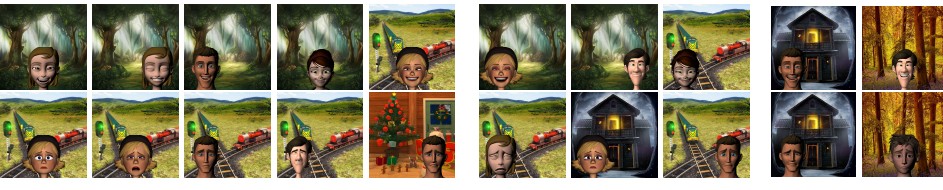

(a) Sample training set      (b) Sample validation set      (c) Sample test set

Figure 1: Example illustration of train/validation/test data. The first row is "happiness" sentiment and the second row is "sadness" sentiment. The background and sentiment labels are correlated in training and validation set, but independent in testing set.

distribution shift. Few would doubt that given photographs of smiling and frowning astronauts on the Martian plains, we could (mostly) agree upon the correct labels.

While we lack a mathematical description of how precisely humans are able to generalize so easily out-of-sample, we can often point to certain classes of perturbations that should not effect the semantics of an image. For example for many tasks, we know that the background should not influence the predictions made about an image. Similarly, other superficial statistics of the data, such as textures or subtle coloring changes should not matter. The essential assumption of this paper is that by making our model depend less on known superficial aspects, we can push the model to rely more on the *difference that makes a difference*. This paper focuses on visual applications, and we focus on high-frequency textural information as the relevant notion of superficial statistics that we *do not* want our model to depend upon.

The contribution of this paper can be summarized as follows.

- We propose a new differentiable neural network building block (neural gray-level co-occurrence matrix) that captures textural information *only* from images without modeling the lower-frequency semantic information that we care about (Section 3.1).

- We propose an architecture-agnostic, parameter-free method that is designed to discard this superficial information, (Section 3.2).

- We introduce two synthetic datasets for DA/DG studies that are more challenging than regular DA/DG scenario in the sense that the domain-specific information is correlated with semantic information. Figure 1 is a toy example (Section 4).

## 2   RELATED WORK IN DOMAIN ADAPTATION AND DOMAIN GENERALIZATION

Domain generalization (DG) (Muandet et al., 2013) is a variation on DA, where samples from the target domain are not available during training. In reality, data-sets may contain data cobbled together from many sources but where those sources are not labeled. For example, a common assumption used to be that there is one and only one distribution for each dataset collected, but Wang et al. (2016) noticed that in video sentiment analysis, the data sources varied considerably even within the same dataset due to heterogeneous data sources and collection practices.

Domain adaptation (Bridle & Cox, 1991; Ben-David et al., 2010a), and (more broadly) transfer learning have been studied for decades, with antecedents in the classic econometrics work on *sample selection bias* Heckman (1977) and *choice models* Manski & Lerman (1977). For a general primer, we refer the reader to these extensive reviews (Weiss et al., 2016; Csurka, 2017).

Domain generalization (Muandet et al., 2013) is relatively new, but has also been studied extensively: covering a wide spectrum of techniques from kernel methods (Muandet et al., 2013; Niu et al., 2015; Erfani et al., 2016; Li et al., 2017c) to more recent deep learning end-to-end methods, where the methods mostly fall into two categories: reducing the inter-domain differences of representations through adversarial (or similar) techniques (Ghifary et al., 2015; Wang et al., 2016; Motiian et al., 2017; Li et al., 2018; Carlucci et al., 2018), or building an ensemble of one-for-each-domain deep

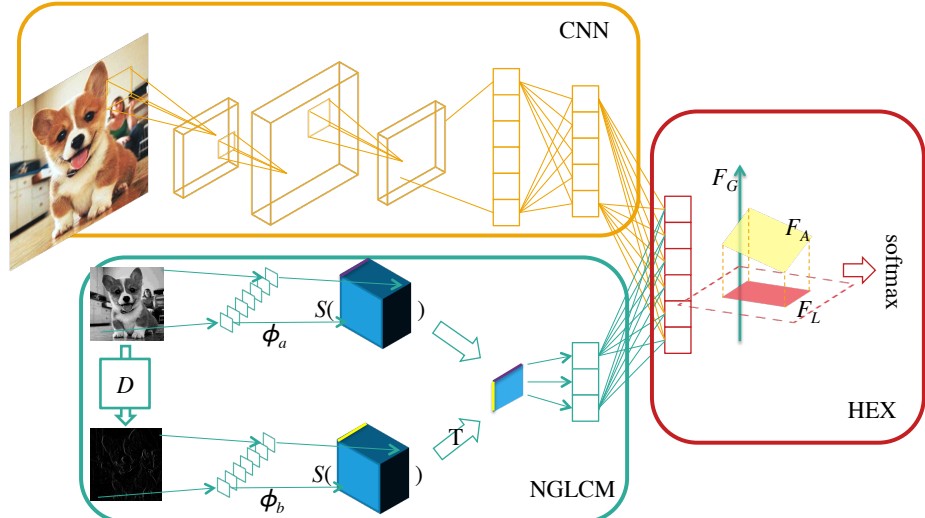

Figure 2: Introduction of Neural Gray-level Co-occurrence Matrix (NGLCM) and HEX.

models and then fusing representations together (Ding & Fu, 2018; Mancini et al., 2018). Meta-learning techniques are also explored (Li et al., 2017b). Related studies are also conducted under the name "zero shot domain adaptation" *e.g.* (Kumagai & Iwata, 2018).

## 3 METHOD

In this section, we introduce our main technical contributions. We will first introduce the our new differentiable neural building block, NGLCM that is designed to capture textural but not semantic information from images, and then introduce our technique for excluding the textural information.

### 3.1 NEURAL GRAY-LEVEL CO-OCCURRENCE MATRIX FOR SUPERFICIAL INFORMATION

Our goal is to design a neural building block that 1) has enough capacity to extract the textural information from an image, 2) is not capable of extracting semantic information. We consulted some classic computer vision techniques for inspiration and extensive experimental evidence (Appendix A1), suggested that *gray-level co-occurrence matrix* (GLCM) (Haralick et al., 1973; Lam, 1996) may suit our goal. The idea of GLCM is to count the number of pixel pairs under a certain direction (common direction choices are $0°$, $45°$, $90°$, and $135°$). For example, for an image $A \in \mathcal{N}^{m \times m}$, where $\mathcal{N}$ denotes the set of all possible pixel values. The GLCM of $A$ under the direction to $0°$ (horizontally right) will be a $|\mathcal{N}| \times |\mathcal{N}|$ matrix (denoted by $G$) defined as following:

$$G_{k,l} = \sum_{i=0}^{m-1} \sum_{j=0}^{m} I(A_{i,j} = k) I(A_{i+1,j} = l) \tag{1}$$

where $|\mathcal{N}|$ stands for the cardinality of $\mathcal{N}$, $I(\cdot)$ is an identity function, $i, j$ are indices of $A$, and $k, l$ are pixel values of $A$ as well as indices of $G$.

We design a new neural network building block that resembles GLCM but whose parameters are differentiable, having (sub)gradient everywhere, and thus are tunable through backpropagation.

We first flatten $A$ into a row vector $a \in \mathcal{N}^{1 \times m^2}$. The first observation we made is that the counting of pixel pairs $(p_k, p_l)$ in Equation 1 is equivalent to counting the pairs $(p_k, \Delta p)$, where $\Delta p = p_k - p_l$. Therefore, we first generate a vector $d$ by multiplying $a$ with a matrix $D$, where $D$ is designed according to the direction of GLCM. For example, $D$ in the $0°$ case will be a $m^2 \times m^2$ matrix $D$ such that $D_{i,i} = 1$, $D_{i,i+1} = -1$, and $0$ elsewhere.

To count the elements in $a$ and $d$ with a differentiable operation, we introduce two sets of parameters $\phi_a \in \mathscr{R}^{|\mathscr{N}| \times 1}$ and $\phi_b \in \mathscr{R}^{|\mathscr{N}| \times 1}$ as the tunable parameter for this building block, so that:

$$G = s(a; \phi_a)s^T(d; \phi_b) \tag{2}$$

where $s()$ is a thresholding function defined as:

$$s(a; \phi_a) = \min(\max(a \ominus \phi_a, 0), 1)$$

where $\ominus$ denotes the minus operation with the broadcasting mechanism, yielding both $s(a; \phi_a)$ and $s(d; \phi_b)$ as $|\mathscr{N}| \times m^2$ matrices. As a result, $G$ is a $|\mathscr{N}| \times |\mathscr{N}|$ matrix.

The design rationale is that, with an extra constrain that requires $\phi$ to have only unique values in the set of $\{n - \epsilon | n \in \mathscr{N}\}$, where $\epsilon$ is a small number, $G$ in Equation 2 will be equivalent to the GLCM extracted with old counting techniques, subject to permutation and scale. Also, all the operations used in the construction of $G$ have (sub)gradient and therefore all the parameters are tunable with backpropagation. In practice, we drop the extra constraint on $\phi$ for simplicity in computation.

Our preliminary experiments suggested that for our purposes it is sufficient to first map standard images with 256 pixel levels to images with 16 pixel levels, which can reduce to the number of parameters of NGLCM ($|\mathscr{N}| = 16$).

## 3.2 HEX

We first introduce notation to represent the neural network. We use $\langle X, y \rangle$ to denote a dataset of inputs $X$ and corresponding labels $y$. We use $h(\cdot; \theta)$ and $f(\cdot; \xi)$ to the bottom and top components of a neural network. A conventional neural network architecture will use $f(h(X; \theta); \xi)$ to generate a corresponding result $F_i$ and then calculate the argmax to yield the prediction label.

Besides conventional $f(h(X; \theta); \xi)$, we introduce another architecture

$$g(X; \phi) = \sigma_m((s(a; \phi_a)s^T(d; \phi_b))W_m + b_m)$$

where $\phi = \{\phi_a, \phi_b, W_m, b_m\}$, $s(a; \phi_a)s^T(d; \phi_b)$ is introduced in previous section, $\{W_m, b_m, \sigma_m\}$ (weights, biases, and activation function) form a standard MLP.

With the introduction of $g(\cdot; \phi)$, the final classification layer turns into $f[h(X; \theta), g(X; \phi)]; \xi]$ (where we use $[\cdot, \cdot]$ to denote concatenation).

Now, with the representation learned through raw data by $h(\cdot; \theta)$ and textural representation learned by $g(\cdot; \phi)$, the next question is to force $f(\cdot; \xi)$ to predict with transformed representation from $h(\cdot; \theta)$ that *in some sense* independent of the superficial representation captured by $g(\cdot; \phi)$.

To illustrate following ideas, we first introduce three different outputs from the final layer:

$$\begin{aligned}
F_A &= f([h(X; \theta), g(X; \phi)]; \xi) \\
F_G &= f([\mathbf{0}, g(X; \phi)]; \xi) \\
F_P &= f([h(X; \theta), \mathbf{0}]; \xi)
\end{aligned} \tag{3}$$

where $F_A$, $F_G$, and $F_P$ stands for the results from both representations (concatenated), only the textural information (prepended with the 0 vector), and only the raw data (concatenated wit hthe 0 vecotr), respectively. $\mathbf{0}$ stands for a padding matrix with all the zeros, whose shape can be inferred by context.

Several heuristics have been proposed to force a network to "forget" some part of a representation, such as adversarial training (Ganin et al., 2016) or information-theoretic regularization (Moyer et al., 2018). In similar spirit, our first proposed solution is to adopt the reverse gradient idea (Ganin et al., 2016) to train $F_P$ to be predictive for the semantic labels $y$ while forcing the $F_P$ to be invariant to $F_G$. Later, we refer to this method as *ADV*. When we use a multilayer perceptron (MLP) to try to predict $g(X; \phi)$ from $h(X; \theta)$ and update the primary model to *fool* the MLP via reverse gradient, we refer to the model as *ADVE*.

Additionally, we introduce a simple alternative. Our idea lies in the fact that, in an affine space, to find a transformation of representation $A$ that is least explainable by some other representation $B$, a straightforward method will be projecting $A$ with a projection matrix constructed by $B$ (sometimes

referred as residual maker matrix.). To utilize this linear property, we choose to work on the space of $F$ generated by $f(\cdot; \xi)$ right before the final argmax function.

Projecting $F_A$ with

$$F_L = (I - F_G(F_G^T F_G)^{-1} F_G^T) F_A \tag{4}$$

will yield $F_L$ for parameter tuning. All the parameters $\xi, \phi, \theta$ can be trained simultaneously (more relevant discussions in Section 5). In testing time, $F_P$ is used.

Due to limited space, we leave the following topics to the Appendix: 1) rationales of this approach (A2.1) 2) what to do in cases when $F_G^T F_G$ is not invertible (A2.2). This method is referred as *HEX*.

Two alternative forms of our algorithm are also worth mentioning: 1) During training, one can tune an extra hyperparameter ($\lambda$) through

$$l(\arg\max F_L, y) + \lambda l(\arg\max F_G, y)$$

to ensure that the NGLCM component is learning superficial representations that are related to the present task where $l(\cdot, \cdot)$ is a generic loss function. 2) During testing, one can use $F_L$, although this requires evaluating the NGLCM component at prediction time and thus is slightly slower. We experimented with these three forms with our synthetic datasets and did not observe significant differences in performance and thus we adopt the fastest method as the main one.

Empirically, we also notice that it is helpful to make sure the textural representation $g(X; \phi)$ and raw data representation $h(X; \theta)$ are of the same scale for HEX to work, so we column-wise normalize these two representations in every minibatch.

## 4 EXPERIMENTS

To show the effectiveness of our proposed method, we conduct range of experiments, evaluating HEX's resilience against dataset shift. To form intuition, we first examine the NGLCM and HEX separately with two basic testings, then we evaluate on two synthetic datasets, on in which *dataset shift* is introduced at the semantic level and another at the raw feature level, respectively. We finally evaluate other two standard domain generalization datasets to compare with the state-of-the-art. All these models are trained with ADAM (Kingma & Ba, 2014).

We conducted ablation tests on our two synthetic datasets with two cases 1) replacing NGLCM with one-layer MLP (denoted as **M**), 2) not using HEX/ADV (training the network with $F_A$ (Equation 3) instead of $F_L$ (Equation 4)) (denoted as **N**). We also experimented with the two alternative forms of HEX: 1) with $F_G$ in the loss and $\lambda = 1$ (referred as HEX-ADV), 2) predicting with $F_L$ (referred as HEX-ALL). We also compare with the popular DG methods (DANN (Ganin et al., 2016)) and another method called information-dropout (Achille & Soatto, 2018).

### 4.1 SYNTHETIC EXPERIMENTS FOR BASIC PERFORMANCE TESTS

#### 4.1.1 NGLCM ONLY EXTRACTS TEXTURAL INFORMATION

To show that the NGLCM only extracts textural information, we trained the network with a mixture of four digit recognition data sets: MNIST (LeCun et al., 1998), SVHN (Netzer et al., 2011), MNIST-M (Ganin & Lempitsky, 2014), and USPS (Denker et al., 1989). We compared NGLCM with a single layer of MLP. The parameters are trained to minimize prediction risk of *digits* (instead of *domain*). We extracted the representations of NGLCM and MLP and used these representations as features to test the five-fold cross-validated Naïve Bayes classifier's accuracy of predicting digit and domain. With two choices of learning rates, we repeated this for every epoch through 100 epochs of training and reported the mean and standard deviation over 100 epochs in Table 1: while MLP and NGLCM perform comparably well in extracting textural information, NGLCM is significantly less useful for recognizing the semantic label.

#### 4.1.2 HEX PROJECTION

To test the effectiveness of HEX, we used the extracted SURF (Bay et al., 2006) features (800 dimension) and GLCM (Lam, 1996) features (256 dimension) from office data set (Saenko et al.,

|        | Random | MLP (1e-2)        | NGLCM (1e-2)      | MLP (1e-4)        | NGLCM (1e-4)      |
|--------|--------|-------------------|-------------------|-------------------|-------------------|
| Domain | 0.25   | $0.686 \pm 0.020$ | $0.738 \pm 0.018$ | $0.750 \pm 0.054$ | $0.687 \pm 0.029$ |
| Label  | 0.1    | $0.447 \pm 0.039$ | $0.161 \pm 0.008$ | $0.534 \pm 0.022$ | $0.142 \pm 0.023$ |

Table 1: Accuracy of domain classification and digit classification

| Train  | Test | Baseline          | HEX               | HEX-ADV           | HEX-ALL           |
|--------|------|-------------------|-------------------|-------------------|-------------------|
| $A, W$ | $D$  | $0.405 \pm 0.016$ | $0.343 \pm 0.030$ | $0.343 \pm 0.030$ | $0.216 \pm 0.119$ |
| $D, W$ | $A$  | $0.112 \pm 0.008$ | $0.147 \pm 0.004$ | $0.147 \pm 0.004$ | $0.055 \pm 0.004$ |
| $A, D$ | $W$  | $0.400 \pm 0.016$ | $0.378 \pm 0.034$ | $0.378 \pm 0.034$ | $0.151 \pm 0.008$ |

Table 2: Accuracy on Office data set with extracted features. The Baseline refers to MLP with SURF features. The HEX methods refer to adding another MLP with features extracted by traditional GLCM methods. Because $D$ and $W$ are similar domains (same obejcts even share the same background), we believe these results favor the HEX method (see Section 4.1.2) for duscussion).

2010) (31 classes). We built a two-layer MLP ($800 \times 256$, and $256 \times 31$) as baseline that only predicts with SURF features. This architecture and corresponding learning rate are picked to make sure the baseline can converge to a relatively high prediction performance. Then we plugged in the GLCM part with an extra first-layer network $256 \times 32$ and the second layer of the baseline is extended to $288 \times 31$ to take in the information from GLCM. Then we train the network again with HEX with the same learning rate.

The Office data set has three different subsets: Webcam ($W$), Amazon ($A$), and DSLR ($D$). We trained and validated the model on a mixture of two and tested on the third one. We ran five experiments and reported the averaged accuracy with standard deviation in Table 2. These performances are not comparable to the state-of-the-art because they are based on features. At first glance, one may frown upon on the performance of HEX because out of three configurations, HEX only outperforms the baseline in the setting $\{W, D\} \rightarrow A$. However, a closer look into the datasets gives some promising indications for HEX: we notice $W$ and $D$ are distributed similarly in the sense that objects have similar backgrounds, while $A$ is distributed distinctly (Appendix A3.1). Therefore, if we assume that there are two classifiers $C_1$ and $C_2$: $C_1$ can classify objects based on object feature and background feature while $C_2$ can only classify objects based on object feature ignoring background feature. $C_2$ will only perform better than $C_1$ in $\{W, D\} \rightarrow A$ case, and will perform worse than $C_2$ in the other two cases, which is exactly what we observe with HEX.

## 4.2 Facial Expression Classification with Nuisance Background

We generated a synthetic data set extending the Facial Expression Research Group Database (Aneja et al., 2016), which is a dataset of six animated individuals expressing seven different sentiments. For each pair of individual and sentiment, there are over $1000$ images. To introduce the data shift, we attach seven different backgrounds to these images. In the training set (50% of the data) and validation set (30% of the data), the background is correlated with the sentiment label with a correlation of $\rho$; in testing set (the rest 20% of the data), the background is independent of the sentiment label. A simpler toy example of the data set is shown in Figure 1. In the experiment, we format the resulting images to $28 \times 28$ grayscale images.

We run the experiments first with the baseline CNN (two convolutional layers and two fully connected layers) to tune for hyperparameters. We chose to run 100 epochs with learning rate 5e-4 because this is when the CNN can converge for all these 10 synthetic datasets. We then tested other methods with the same learning rate. The results are shown in Figure 3 with testing accuracy and standard deviation from five repeated experiments. Testing accuracy is reported by the model with the highest validation score. In the figure, we compare baseline CNN (**B**), Ablation Tests (**M** and **N**), ADV (**A**), HEX (**H**), DANN (**G**), and InfoDropout (**I**). Most these methods perform well when $\rho$ is small (when testing distributions are relatively similar to training distribution). As $\rho$ increases, most methods' performances decrease, but Adv and HEX behave relatively stable across these ten correlation settings. We also notice that, as the correlation becomes stronger, **M** deteriorates at a faster pace than other methods. Intuitively, we believe this is because the MLP learns both from the *seman-*

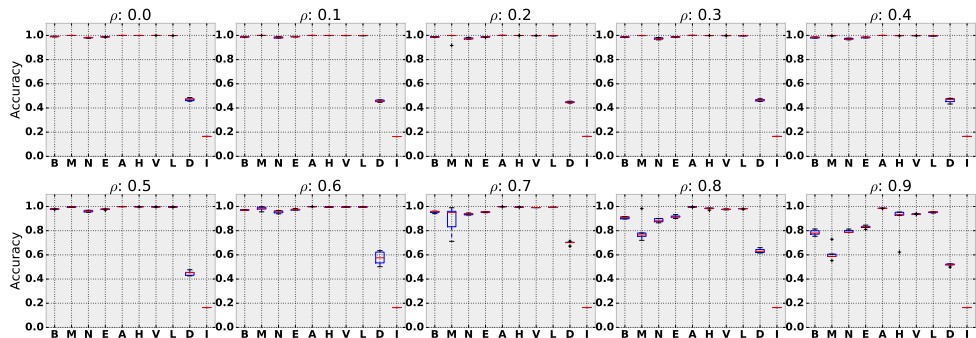

Figure 3: Averaged testing accuracy and standard deviation of five repeated experiments with different correlation level on sentiment with nuisance background data. Notations: baseline CNN (**B**), Ablation Tests (**M** (replacing NGLCM with MLP) and **N** (training without HEX projection)), ADVE (**E**), ADV (**A**), HEX (**H**), HEX-ADV (**V**), HEX-ALL (**L**), DANN (**G**), and InfoDropout (**I**).

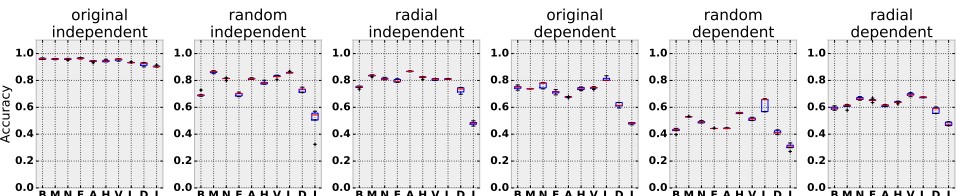

Figure 4: Averaged testing accuracy and standard deviation of five repeated experiments with different strategies of attaching patterns to MNIST data. Notations: baseline CNN (**B**), Ablation Tests (**M** (replacing NGLCM with MLP) and **N** (training without HEX projection)), ADVE (**E**), ADV (**A**), HEX (**H**), HEX-ADV (**V**), HEX-ALL (**L**), DANN (**G**), and InfoDropout (**I**).

*tic* signal together with superficial signal, leading to inferior performance when HEX projects this signal out. We also notice that ADV and HEX improve the speed of convergence (Appendix A3.2).

### 4.3 Mitigating the Tendency of Surface Statistical Regularities in MNIST

As Jo & Bengio (2017) observed, CNNs have a tendency to learn the surface statistical regularities: the generalization of CNNs is partially due to the abstraction of high level semantics of an image, and partially due to surface statistical regularities. Here, we demonstrate the ability of HEX to overcome such tendencies. We followed the radial and random Fourier filtering introduced in (Jo & Bengio, 2017) to attach the surface statistical regularities into the images in MNIST. There are three different regularities altogether (radial kernel, random kernel, and original image). We attached two of these into training and validation images and the remaining one into testing images. We also adopted two strategies in attaching surface patterns to training/validation images: 1) *independently*: the pattern is independent of the digit, and 2) *dependently*: images of digit 0-4 have one pattern while images of digit 5-9 have the other pattern. Some examples of this synthetic data are shown in Appendix A3.3.

We used the same learning rate scheduling strategy as in the previous experiment. The results are shown in Figure 4. Figure legends are the same as previous. Interestingly, NGLCM and HEX contribute differently across these cases. When the patterns are attached independently, **M** performs the best overall, but when the patterns are attached dependently, **N** and HEX perform the best overall. In the most challenging case of these experiments (random kerneled as testing, pattern attached dependently), HEX shows a clear advantage. Also, HEX behaves relatively more stable overall.

| Test | CAE | MTAE | CCSA | DANN | Fusion | LabelGrad | CrossGrad | HEX | ADV |
|------|-----|------|------|------|--------|-----------|-----------|-----|-----|
| $\mathcal{M}_{0°}$ | 72.1 | 82.5 | 84.6 | 86.7 | 85.6 | 89.7 | 88.3 | **90.1** | **89.9** |
| $\mathcal{M}_{15°}$ | 95.3 | 96.3 | 95.6 | 98 | 95.0 | 97.8 | 98.6 | **98.9** | 98.6 |
| $\mathcal{M}_{30°}$ | 92.6 | 93.4 | 94.6 | 97.8 | 95.6 | 98.0 | 98.0 | **98.9** | 98.8 |
| $\mathcal{M}_{45°}$ | 81.5 | 78.6 | 82.9 | 97.4 | 95.5 | 97.1 | 97.7 | **98.8** | 98.7 |
| $\mathcal{M}_{60°}$ | 92.7 | 94.2 | 94.8 | 96.9 | 95.9 | 96.6 | 97.7 | 98.3 | **98.6** |
| $\mathcal{M}_{75°}$ | 79.3 | 80.5 | 82.1 | 89.1 | 84.3 | **92.1** | 91.4 | 90.0 | 90.4 |
| Avg | 85.6 | 87.6 | 89.1 | 94.3 | 92.0 | 95.2 | 95.3 | **95.8** | 95.2 |

Table 3: Accuracy on MNIST-Rotation data set

## 4.4 MNIST WITH ROTATION AS DOMAIN

We continue to compare HEX with other state-of-the-art DG methods (that use distribution labels) on popular DG data sets. We experimented with the MNIST-rotation data set, on which many DG methods have been tested. The images are rotated with different degrees to create different domains. We followed the approach introduced by Ghifary et al. (2015). To reiterate: we randomly sampled a set $\mathcal{M}$ of 1000 images out of MNIST (100 for each label). Then we rotated the images in $\mathcal{M}$ counter-clockwise with different degrees to create data in other domains, denoted by $\mathcal{M}_{15°}$, $\mathcal{M}_{30°}$, $\mathcal{M}_{45°}$, $\mathcal{M}_{60°}$, $\mathcal{M}_{75°}$. With the original set, denoted by $\mathcal{M}_{0°}$, there are six domains altogether.

We compared the performance of HEX/ADV with several methods tested on this data including CAE (Rifai et al., 2011), MTAE (Ghifary et al., 2015), CCSA (Motiian et al., 2017), DANN (Ganin et al., 2016), Fusion (Mancini et al., 2018), LabelGrad, and CrossGrad (Shankar et al., 2018). The results are shown in Table 3: HEX is only inferior to previous methods in one case and leads the average performance overall.

## 4.5 PACS: GENERALIZATION IN PHOTO, ART, CARTOON, AND SKETCH

Finally, we tested on the PACS data set (Li et al., 2017a), which consists of collections of images of seven different objects over four domains, including photo, art painting, cartoon, and sketch.

Following (Li et al., 2017a), we used AlexNet as baseline method and built HEX upon it. We met some optimization difficulties in directly training AlexNet on PACS data set with HEX, so we used a heuristic training approach: we first fine-tuned the AlexNet pretrained on ImageNet with PACS data of training domains without plugging in NGLCM and HEX, then we used HEX and NGLCM to further train the top classifier of AlexNet while the weights of the bottom layer are fixed. Our heuristic training procedure allows us to tune the AlexNet with only 10 epoches and train the top-layer classifier 100 epochs (roughly only 600 seconds on our server for each testing case).

We compared HEX/ADV with the following methods that have been tested on PACS: AlexNet (directly fine-tuning pretrained AlexNet on PACS training data (Li et al., 2017a)), DSN (Bousmalis et al., 2016), L-CNN (Li et al., 2017a), MLDG (Li et al., 2017b), Fusion (Mancini et al., 2018). Notice that most of the competing methods (DSN, L-CNN, MLDG, and Fusion) have explicit knowledge about the domain identification of the training images. The results are shown in Table 4. Impressively, HEX is only slightly shy of Fusion in terms of overall performance. Fusion is a method that involves three different AlexNets, one for each training domain, and a fusion layer to combine the representation for prediction. The Fusion model is roughly three times bigger than HEX since the extra NGLCM component used by HEX is negligible in comparison to AlexNet in terms of model complexity. Interestingly, HEX achieves impressively high performance when the testing domain is Art painting and Cartoon, while Fusion is good at prediction for Photo and Sketch.

## 5 DISCUSSION AND CONCLUSION

We introduced two novel components: NGLCM that only extracts textural information from an image, and HEX that projects the textural information out and forces the model to focus on semantic information. Limitations still exist. For example, NGLCM cannot be completely free of semantic information of an image. As a result, if we apply our method on standard MNIST data set, we will

| Test Domain | AlexNet | DSN | L-CNN | MLDG | Fusion | HEX | ADV |
|---|---|---|---|---|---|---|---|
| Art | 63.3 | 61.1 | 62.8 | 63.6 | 64.1 | **66.8** | 64.9 |
| Cartoon | 63.1 | 66.5 | 66.9 | 63.4 | 66.8 | **69.7** | 69.6 |
| Photo | 87.7 | 83.2 | 89.5 | 87.8 | **90.2** | 87.9 | 88.2 |
| Sketch | 54 | 58.5 | 57.5 | 54.9 | **60.1** | 56.3 | 55.5 |
| Average | 67.0 | 67.3 | 69.2 | 67.4 | **70.3** | 70.2 | 69.5 |

Table 4: Testing Accuracy on PACS

see slight drop of performance because NGLCM also learns some semantic information, which is then projected out. Also, training all the model parameters simultaneously may lead into a trivial solution where $F_G$ (in Equation 3) learns garbage information and HEX degenerates to the baseline model. To overcome these limitations, we invented several training heuristics, such as optimizing $F_P$ and $F_G$ sequentially and then fix some weights. However, we did not report results with training heuristics (expect for PACS experiment) because we hope to simplify the methods. Another limitation we observe is that sometimes the training performance of HEX fluctuates dramatically during training, but fortunately, the model picked up by highest validation accuracy generally performs better than competing methods. Despite these limitations, we still achieved impressive performance on both synthetic and popular DG data sets.

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

APPENDIX

## A1 REASONS TO CHOOSE GLCM

In order to search the old computer vision techniques for a method that can extract more textural information and less semantic information, we experimented with three classifcal computer vision techniques: SURF (Bay et al., 2006), LBP (He & Wang, 1990), and GLCM (Haralick et al., 1973) on several different data sets: 1) a mixture of four digit data sets (MNIST (LeCun et al., 1998), SVHN (Netzer et al., 2011), MNIST-M (Ganin & Lempitsky, 2014), and USPS (Denker et al., 1989)) where the semantic task is to recognize the digit and the textural task is to classify the which data set the image is from; 2) a rotated MNIST data set with 10 different rotations where the semantic task is to recognize the digit and the textural task is to classify the degrees of rotation; 3) a MNIST data randomly attached one of 10 different types of radial kernel, for which the semantic task is to recognize digits and the textural task is to classify the different kernels.

|  |  | LBP | SURF | GLCM |
|---|---|---|---|---|
| Digits | Semantic | 0.179 | 0.563 | 0.164 |
|  | Textural | 0.527 | 0.809 | 0.952 |
| Rotated Digit | Semantic | 0.155 | 0.707 | 0.214 |
|  | Textural | 0.121 | 0.231 | 0.267 |
| Kernelled | Semantic | 0.710 | 0.620 | 0.220 |
|  | Textural | 0.550 | 0.200 | 0.490 |

Table A1: Accuracy in classifying semantic and superficial information

From results in Table A1, we can see that GLCM suits our goal very well: GLCM outperforms other methods in most cases in classifying textural patterns while predicts least well in the semantic tasks.

## A2 EXPLANATION OF HEX

### A2.1 MATHEMATICAL RATIONALE

With $F_A$ and $F_G$ calculated in Equation 3, we need to transform the representation of $F_A$ so that it is least explainable by $F_G$. Directly adopting subtraction maybe problematic because the $F_A - F_G$ can still be correlated with $F_G$. A straightforward way is to regress the information of $F_G$ out of $F_A$. Since both $F_A$ and $F_G$ are in the same space and the only operation left in the network is the argmax operation, which is linear, we can safely use linear operations.

To form a standard linear regression problem, we first consider the column $k$ of $F_A$, denoted by $F_A^{(k)}$. To solve a standard linear regression problem is to solve:

$$\beta^{\hat{(k)}} = \arg\min_{\beta^{(k)}} ||F_A^{(k)} - F_G\beta^{(k)}||_2^2$$

This function has a closed form solution when the minibatch size is greater than the number of classes of the problem (*i.e.* when the number of rows of $F_G$ is greater than number of columns of $F_G$), and the closed form solution is:

$$\beta^{\hat{(k)}} = \frac{F_G^T F_A^{(k)}}{(F_G^T F_G)}$$

Therefore, for $k^{\text{th}}$ column of $F_A$, what cannot be explained by $F_G$ is (denoted by $F_L^{(k)}$):

$$F_L^{(k)} = F_A^{(k)} - F_G\frac{F_G^T F_A^{(k)}}{(F_G^T F_G)} = (I - F_G(F_G^T F_G)^{-1}F_G^T)F_A^{(k)}$$

Repeat this for every column of $F_A$ will lead to:

$$F_L = (I - F_G(F_G^T F_G)^{-1}F_G^T)F_A$$

which is Equation 4.

## A2.2 When $F_G^T F_G$ is not invertible

As we mentioned above, Equation 4 can only be derived when the minibatch size is greater than the number of classes to predict because $F_G^T F_G$ is only non-singular (invertible) when this condition is met.

Therefore, a simple technique to always guarantee a solution with HEX is to use a minibatch size that is greater than the number of classes. We believe this is a realistic requirement because in the real-world application, we always know the number of classes to classify, and it is usually a number much smaller than the maximum minibatch size a modern computer can deal with.

However, to complete this paper, we also introduce a more robust method that is always applicable independent of the choices of minibatch sizes.

We start with the simple intuition that to make sure $F_G^T F_G$ is always invertible, the simplest conduct will be adding a smaller number to the diagonal, leading to $F_G^T F_G + \lambda I$, where we can end the discussion by simply treating $\lambda$ as a tunable hyperparameter.

However, we prefer that our algorithm not require tuning additional hyperparameters. We write $F_G^T F_G + \lambda I$ back to the previous equation,

$$F_L^{(k)} = (I - F_G(F_G^T F_G + \lambda I)^{-1} F_G^T) F_A^{(k)}$$

With the Kailath Variant (Bishop et al., 1995), we can have:

$$F_L^{(k)} = F_A^{(k)} - F_G \frac{F_G^T (F_G F_G^T + \lambda I)^{-1} F_A^{(k)}}{F_G^T (F_G F_G^T + \lambda I)^{-1} F_G} = F_A^{(k)} - F_G \beta_\lambda^{(k)}$$

where $\beta_\lambda^{(k)}$ is a result of a heteroscadestic regression method where $\lambda$ can be estimated through maximum likelihood estimation (MLE) (Wang et al., 2017), which completes the story of a hyperparameter-free method even when $F_G^T F_G$ is not invertible.

However, in practice, we notice that the MLE procedure is very slow and the estimation is usually sensitive to noise. As a result, we recommend users to simply choose a larger minibatch size to avoid the problem. Nonetheless, we still release these steps here to 1) make the paper more complete, 2) offer a solution when in rase cases a model is asked to predict over hundreds or thousands of classes. Also, we name our main method "HEX" as short of heteroscadestic regression.

## A3 Extra Experiment Results

### A3.1 A closer look into Office Data set

We visualize some images of the office data set in Figure A1, where we can see that the background of images for DSLR and Webcam are very similar while the background of images in Amazon are distinctly different from these two.

### A3.2 HEX converges much faster

We plotted the testing accuracy of each method in the facial expression classification in Figure A2. From the figure, we can see that HEX and related ablation methods converge significantly faster than baseline methods.

### A3.3 Examples of Pattern-attached MNIST data set

Examples of MNIST images when attached with different kernelled patterns following (Jo & Bengio, 2017), as shown in Figure A3.

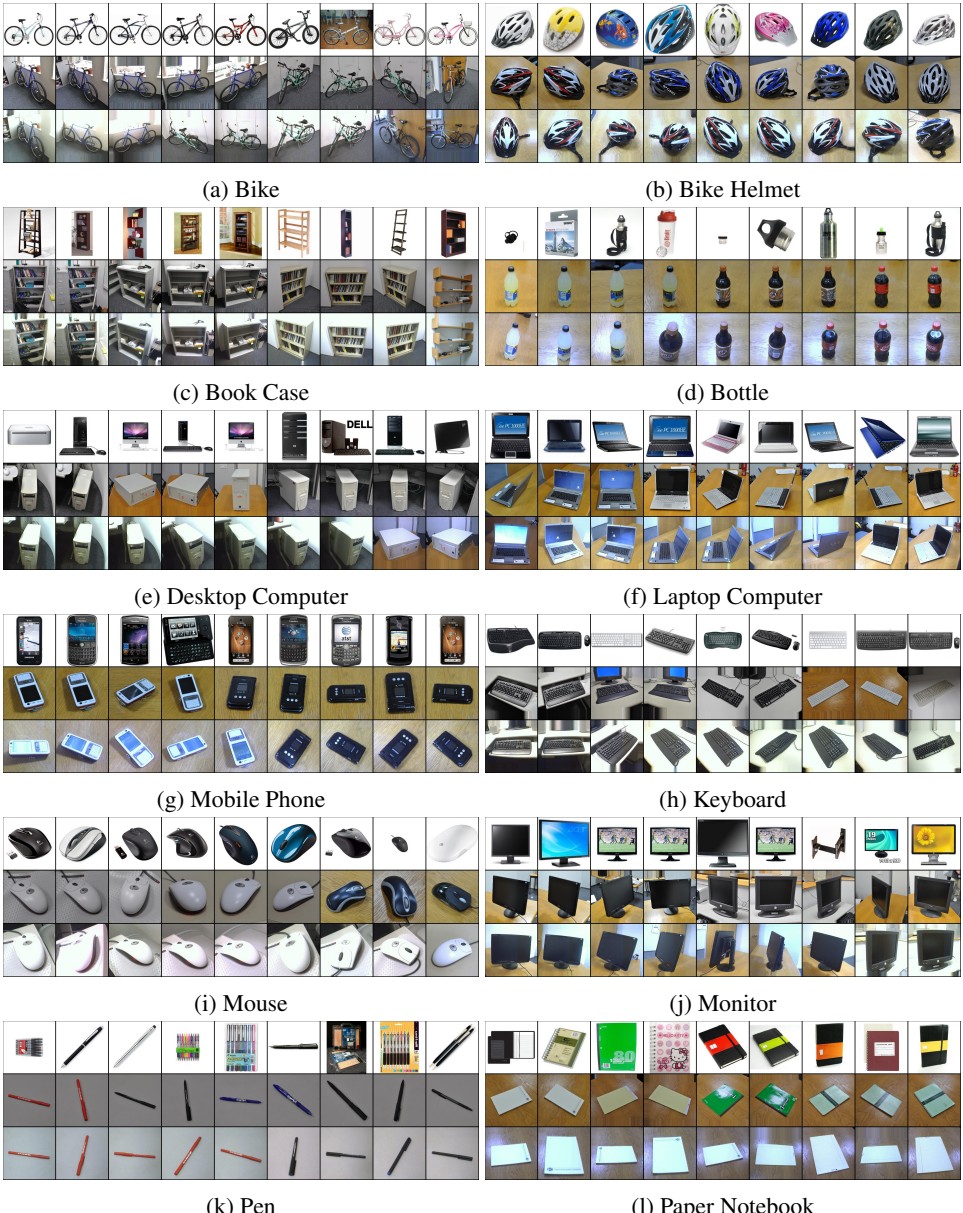

(a) Bike

(b) Bike Helmet

(c) Book Case

(d) Bottle

(e) Desktop Computer

(f) Laptop Computer

(g) Mobile Phone

(h) Keyboard

(i) Mouse

(j) Monitor

(k) Pen

(l) Paper Notebook

Figure A1: A closer look of Office data set, we visualize the first 10 images of each data set. We show 12 labels out of 31 labels, but the story of the rest labels are similar to what we have shown here. From the images, we can clearly see that many images of DSLR and Webcam share the similar background, while the images of Amazon have a distinct background. Top row: Amazon, middle row: DSLR, bottom row: Webcam

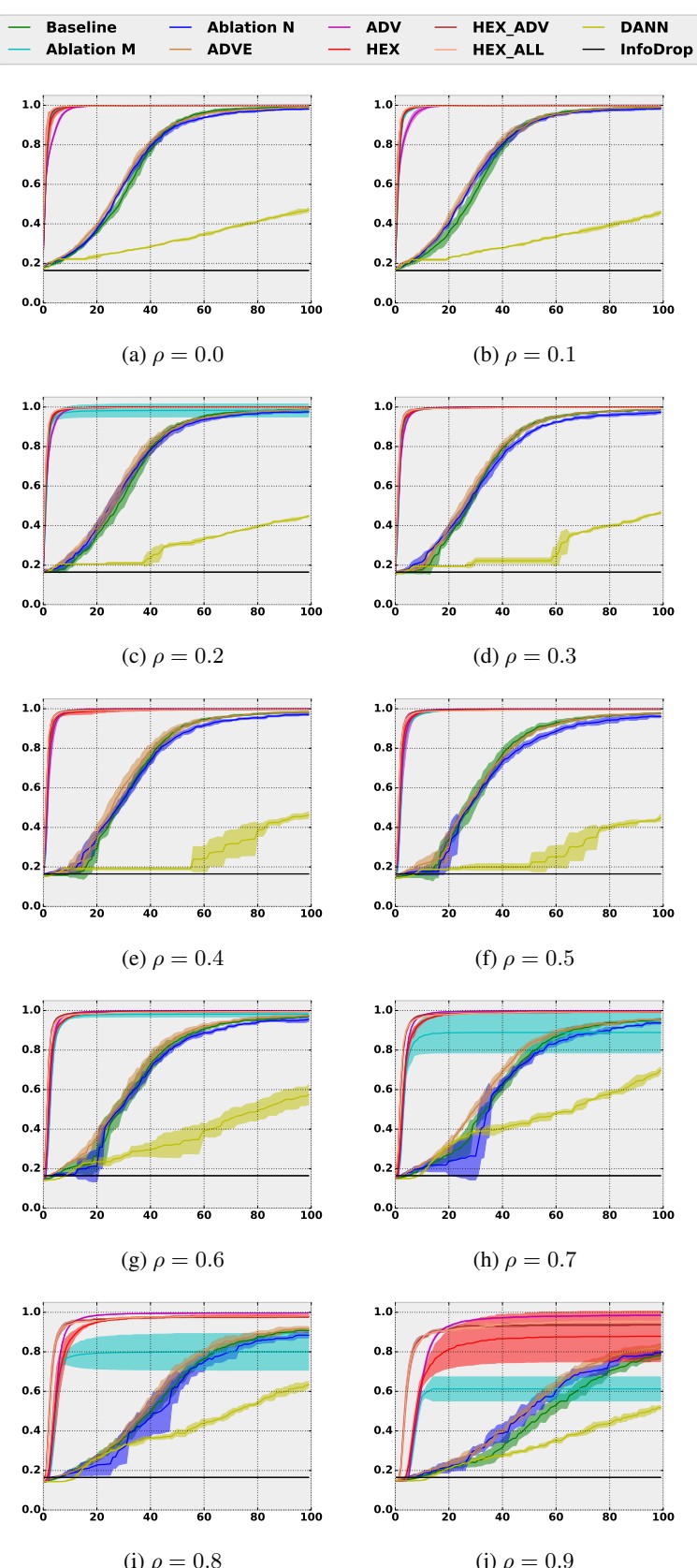

Figure A2: Testing accuracy curve of the facial expression classification experiment.

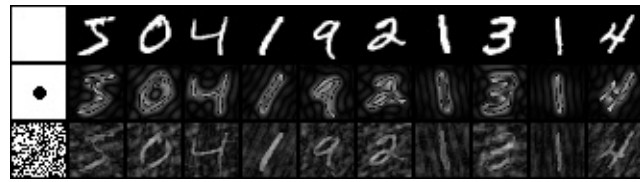

Figure A3: Synthetic MNIST data sets with Fourier transform patterns. The leftmost image represents the kernel.

