# OpenReview forum: "Learning Robust Representations by Projecting Superficial Statistics Out"
_ICLR.cc/2019/Conference_

### Official Review · AnonReviewer3 · 2018-10-31
**The proposal is well structured and written. The quality of the paper is excellent in terms of novelty and originality.**

**Rating:** 9
**Confidence:** 3

**Review:**

The paper proposed a novel differentiable neural GLCM network which captures the high reference textural information and discard the lower-frequency semantic information so as to solve the domain generalisation challenge. The author also proposed an approach “HEX” to discard the superficial representations. Two synthetic datasets are created for demonstrating the methods advantages on scenarios where the domain-specific information is correlated with the semantic information. The proposal is well structured and written. The quality of the paper is excellent in terms of novelty and originality. The proposed methods are evaluated thoroughly through experiments with different types of dataset and has shown to achieve good performance.

---

> ### Author Response · Authors · 2018-11-21
> **Reply to Reviewer 3**
>
> Thank you for the strong positive assessment of our work. We’re glad that you appreciated the originality of our approach, the value of our new datasets, and the quality of our exposition. We will continue to improve the draft in the camera-ready version.

---

### Official Review · AnonReviewer2 · 2018-11-01
**Revisit old image processing idea, add parameters, make differentiable. Show that it can be used to ignore background textures. Extensive experiments on domain adaptation.**

**Rating:** 7
**Confidence:** 4

**Review:**

Summary:
The paper proposes an unsupervised approach to identify image features that are not meaningful for image classification tasks. The goal is to address the domain adaptation (DA)/domain generalization (DG) issue. The paper introduces a new learning task where the domain identity is unavailable during training, called unguided domain generalization (UDG). The proposed approach is based on an old method of using gray level co-occurence matrix, updated to allow for differentiable training. This new approach is used in two different ways to reduce the effect of background texture in a classification task. The paper introduces a new dataset, and shows extensive and carefully designed experiments using the new data as well as existing domain generalization datasets.

This paper revisits an old idea from image processing in a new way, and provides an interesting unsupervised method for identifying so called superficial features. The proposed block seems to be very modular in design, and can be plugged into other architectures. The main weakness is that it is a bit unclear exactly what is being assumed as "background texture" by the authors.


Overall comments:
- Some more clarity on what you mean by superficial statistics would be good. E.g. by drawing samples. Are you assuming the object is centered? Somehow filling the image?  Different patch statistics? How about a texture classification task?
- please derive why NGLCM reduces to GLCM in the appendix. Also show the effect of dropping the uniqueness constraint.
- Section 3.2: I assume you are referring to an autoencoder style architecture here. Please rewrite the first paragraph. The current setup seems to indicate that you doing supervised training, since you have labels y, but then you talk about decoder and encoder.
- Section 3.2: Please expand upon why you use F_L for training but F_P during testing


Minor typos/issues:
- Last bullet in Section 1: DG not yet defined, only defined in Section 2.
- page 2, Section 2, para 1: data collection conduct. Please reword.
- page 2, Section 2, para 2: Sentence: For a machine learning ... There is no object in this sentence. Not sure what you are trying to define.
- page 2, Section 2, para 2: Is $\mathcal{S}$ and $\mathcal{T}$ not intersecting?
- page 2, Section 2.1: Heckman (1977), use \citep
- page 2, Section 2.1: Manski, citep and missing year
- page 3, Section 2.1: Kumagai, use citet
- page 3, Section 3.1: We first expand ... --> We first flatten A into a row vector
- page 4, Section 3.1: b is undefined. I assume you mean d?
- page 4, Section 3.1: twice: contrain --> constraint
- page 4, Section 3.2: <X,y> --> {X,y} as used in Section 3.1.
- page 4, Section 3.2, just below equation: as is introduced in the previous section. New sentence about MLP please. And MLP not defined.
- page 4, Section 3.2, next paragraph: missing left bracket (
- page 4, Section 3.2: inferred from its context.
- page 5, Section 4: popular DG method (DANN)
- page 7: the rest one into --> the remaining one into
- page 8: rewrite: when the empirical performance interestingly preserves.
- page 8, last sentence: GD --> DG
- A2.2: can bare with. --> can deal with.
- A2.2: linear algebra and Kailath Variant. Unsure what you are trying to say.
- A2.2: sensitive to noises --> sensitive to noise.

---

> ### Author Response · Authors · 2018-11-21
> **Reply to Reviewer 2**
>
> Thanks for a detailed review. We are grateful both for your big-picture feedback and for your extensive granular suggestions to improve the exposition of our paper. We were glad to see that you appreciated our creativity in using GLCM and recognized the modularity of our design. Your question regarding F_L and F_P is insightful and we’re glad that you identified this missing detail in the paper. We compared evaluation with F_L and F_P and discovered that performance was equivocal. This favors the use of F_P, allowing us to use the machinery of the GLCM at training time but discarding it at test time. We promise to add this discussion and supporting experiments to the camera-ready version. Additionally, we will revise the first paragraph of 3.2 per your suggestions and fix the numerous small typos and type-setting corrections that you identified. Thanks again for your generous feedback and attention to detail.

---

### Official Review · AnonReviewer1 · 2018-11-04
**A domain generalization approach is introduced to reveal semantic (relevant) information based on a linear projection scheme from CNN and NGLCM ouput layers.**

**Rating:** 7
**Confidence:** 4

**Review:**

The paper is clear regarding motivation, related work, and mathematical foundations. The introduced cross-local intrinsic dimensionality- (CLID) seems to be naive but practical for GAN assessment. Notably, the experimental results seem to be convincing and illustrative.

The domain generalization idea from CNN-based discriminative feature extraction and gray level co-occurrence matrix-based high-frequency coding (superficial information), is an elegant strategy to favor domain generalization. Indeed, the linear projection learned from CNN, and GLCM features could be extended to different real-world applications regarding domain generalization and transferring learning. So, the paper is clear to follow and provides significant insights into a current topic.

Pros:
- Clear mathematical foundations.
- The approach can be applied to different up-to-date problems.
-Though the obtained results are fair, the introduced approach would lead to significant breakthroughs regarding domain generalization techniques.

Cons:
-Some experimental results can be difficult to reproduce. Indeed, authors claim that the training heuristic must be enhanced.
-Table 2 results are not convincing.

---

> ### Author Response · Authors · 2018-11-21
> **Reply to Reviewer 1**
>
> Thank you very much for these comments. We are glad that you appreciated the paper’s overall aims and recognized the general applicability of the methodology that we propose. We are also grateful for your constructive suggestions:
>
> * To address your concerns about reproducibility we will add an appendix providing extensive detail about all heuristics employed during training. Additionally, we plan to release open source version of all of our code upon publication.
>
> * Regarding Table 2: thanks for pointing this out. We agree that while an argument can be found in the main text, Table 2 is poorly described in the caption and must be better presented in the camera-ready version. In short, domains D and W here overlap significantly. Therefore a model trained on one and evaluated on the other perform well, and we conjectured that discarding the superficial information can actually degrade performance.

---

### Author Response · Authors · 2018-11-21
**General Reply to Reviews**

We would like to thank all of the reviewers for their constructive reviews. Overall, we are glad to see that all three reviewers champion the paper, appreciating the paper’s overall aim, creativity in revisiting GLCM, proposed experiment set-ups, and the strength of the empirical results. We are also grateful for the reviewers’ constructive suggestions which will help to improve the camera-ready version of the paper. Please find comments We will answer the reviewers’ comments individually.

---

### Meta-Review · Area_Chair1 · 2018-12-11
**Original work for domain generalization with strong experimental evidence**

**Confidence:** 4
**Recommendation:** Accept (Oral)

**Metareview:**

The paper presents a new approach for domain generalization whereby the original supervised model is trained with an explicit objective to ignore so called superficial statistics present in the training set but which may not be present in future test sets. The paper proposes using a differentiable variant of gray-level co-occurrence matrix to capture the textural information and then experiments with two techniques for learning feature invariance. All reviewers agree the approach is novel, unique, and potentially high impact to the community.

The main issues center around reproducibility as well as the intended scope of problems this approach addresses. The authors have offered to include further discussions in the final version to address these points. Doing so will strengthen the paper and aid the community in building upon this work.